# The DJ-1-Binding Compound Exerts a Protective Effect in Both In Vitro and In Vivo Models of Sepsis-Induced Acute Kidney Injury

**DOI:** 10.3390/antiox14060719

**Published:** 2025-06-12

**Authors:** Réka Zrufkó, Csenge Pajtók, Beáta Szebeni, Apor Veres-Székely, Mária Bernáth, Csenge Szász, Péter Bokrossy, Attila J. Szabó, Ádám Vannay, Domonkos Pap

**Affiliations:** 1Pediatric Center, MTA Center of Excellence, Semmelweis University, 1083 Budapest, Hungary; zrufko.reka@phd.semmelweis.hu (R.Z.); pajtok.csenge@gmail.com (C.P.); szebeni.beata@semmelweis.hu (B.S.); veres-szekely.apor@semmelweis.hu (A.V.-S.); bernath.maria@gmail.com (M.B.); szasz.csenge@phd.semmelweis.hu (C.S.); bokrossy.peter@phd.semmelweis.hu (P.B.); szabo.attila@semmelweis.hu (A.J.S.); vannay.adam@semmelweis.hu (Á.V.); 2HUN-REN-SU Pediatrics and Nephrology Research Group, 1052 Budapest, Hungary

**Keywords:** DJ-1, PARK7, sepsis, acute kidney injury, oxidative stress, inflammation, antioxidant, drug development

## Abstract

Although sepsis-induced acute kidney injury (AKI) is associated with significant morbidity and mortality, its treatment remains unresolved. Oxidative stress and inflammation are key elements in the pathomechanism of AKI. Therefore, in the present study, we investigated the role of DJ-1 protein, known for its antioxidant and anti-inflammatory properties in an animal model of lipopolysaccharide (LPS)-induced AKI. The presence of DJ-1 was detected by immunofluorescence staining in mice kidney samples, human embryonic kidney cells (HEK-293), and peripheral blood mononuclear cells (PBMCs). To investigate DJ-1 functions, Compound-23, a specific DJ-1-binding and preserving compound (CAS: 724737-74-0), was used in vitro and in vivo. Compound-23 reduced the H_2_O_2_-induced reactive oxygen species (ROS) production of the HEK-293 cells, and their LPS- or H_2_O_2_-induced death, as well. In accordance, Compound-23 decreased the mRNA expression of the oxidative stress markers NAD(P)H quinone dehydrogenase 1 (*NQO1*) and glutamate-cysteine ligase (*GCLC*) in the LPS-treated, and *NQO1* in the H_2_O_2_-treated cells. Moreover, Compound-23 reduced the H_2_O_2_- and LPS-induced mRNA expression of inflammatory cytokine interleukin 6 (*IL6*) in both HEK-293 and PBMCs. Using the mice model of LPS-induced AKI, we demonstrated that Compound-23 treatment improved the renal functions of the mice. In addition, Compound-23 decreased the renal mRNA expression of kidney injury molecule 1 (*Kim1*), neutrophil gelatinase-associated lipocalin (*Ngal*), *Nqo1*, *Gclc*, and *Il6* in the LPS-treated mice. Our study revealed that compounds protecting DJ-1 functions may protect the kidney from LPS-induced damage, suggesting that DJ-1 could be a potential drug target for sepsis-induced AKI therapy.

## 1. Introduction

Sepsis is a life-threatening condition characterized by dysregulated immune response to infection, often leading to organ failure. Among complications, acute kidney injury (AKI) is one of the most serious, affecting up to 50% of critically ill patients with sepsis [1]. AKI is associated with significant morbidity and mortality and poses long-term risks of progression to chronic kidney disease (CKD) [2]. The disease pathophysiology is multifactorial and involves a combination of inflammation and oxidative stress [3]. These processes interact synergistically to impair kidney functions. Despite the huge medical burden, there is no specific treatment against sepsis-induced AKI.

Protein Deglycase DJ-1, also known as Parkinson’s disease protein 7 (PARK7), is a small, homodimer protein that was primarily studied regarding the central nervous system and cancer diseases [4,5]. DJ-1 is considered a ubiquitously expressed protein in various tissues, including the kidney. Indeed, its expression in proximal tubular epithelial cells and podocytes was described. In accordance with this, in this study, we also demonstrated its renal expression in the glomerular and tubular cells (Figure 1). Recent studies suggested involvement of DJ-1 in renal diseases. Indeed, overexpression of DJ-1 was demonstrated to attenuate oxidative-stress-induced damage of renal epithelial cells [6]. In addition, the protective effect of DJ-1 was described in autosomal dominant polycystic kidney diseases [7]. Also, the study of De Miguel et al. found that ND-13, a DJ-1 derived peptide, decreases unilateral ureter-obstruction-induced renal fibrosis [8]. However, our knowledge about the role of DJ-1 in sepsis-associated AKI is limited.

The vast majority of the literature data describe DJ-1 as a key player of antioxidant defense and its anti-inflammatory and chaperone activity have also been described [9,10,11]. DJ-1 can eliminate reactive oxygen species (ROS) through various mechanisms, including the stabilization of mitochondrial complexes, scavenging of reactive oxygen radicals via its redox-sensitive cysteine residues, and activation of antioxidant response genes—such as γ-glutamylcysteine synthetase (GCLC) and NAD(P)H quinone oxidoreductase 1 (NQO1)—through the NF-E2-related factor 2 (NRF2) pathway [12,13,14,15]. DJ-1 is sensitive to oxidative stress and the excessive oxidation of its C106 residue renders the protein inactive [16,17,18]. Compound 23 (Compound-23, CAS: 724737-74-0), developed by Kitamura et al., specifically binds to the C106 region of DJ-1, thereby inhibiting excessive oxidation of this chaperon and stabilizing its protein structure, which enhances its antioxidant and protective functions [19,20]. In accordance, our research group and others demonstrated the protective effect of Compound-23 against oxidative-stress-induced cell death on intestinal epithelial or neuronal cells [20,21]. Moreover, it was also published that DJ-1 deficiency led to increased pulmonary inflammation and oxidative stress in the mouse model of lipopolysaccharide (LPS)-induced acute lung injury [22]

Since oxidative stress and inflammation are the hallmarks of sepsis-induced AKI, in this study, we investigated whether the pharmacological protection of DJ-1 with Compound-23 may have therapeutic potential in the disease. To this end, we examined the expression of DJ-1 in kidney samples, human embryonic kidney cells (HEK-293) and peripheral blood mononuclear cells (PBMCs). Thereafter, we investigated the in vitro effects of Compound-23 on reactive oxygen species production, apoptosis, DJ-1-related antioxidant gene expression (GCLC, NQO1), and inflammatory cytokine (IL-6) expression in HEK-293 cells and PBMCs exposed to bacterial lipopolysaccharide (LPS) or oxidative stress. Finally, in accordance with the in vitro experiments, we examined the effect of Compound-23 in the animal model of LPS-induced acute kidney injury on renal damage severity, functional decline, and the expression of DJ-1-mediated antioxidant genes and inflammatory genes, as well. These measurements not only help to elucidate the role of DJ-1 in the pathomechanism of sepsis-induced acute kidney injury (AKI), but also highlight whether compounds that protect DJ-1—such as Compound-23—may be applicable in the therapeutic management of sepsis-induced AKI.

## 2. Materials and Methods

### 2.1. Cell Lines

Human embryonic kidney 293 (HEK-293) (American Type Culture Collection, Manassas, VA, USA) cells were cultured in Dulbecco’s Modified Eagle Medium (DMEM) (Thermo Fisher Scientific, Waltham, MA, USA) medium, supplemented with 10% heat-inactivated fetal bovine serum (FBS, Thermo Fisher Scientific, Waltham, MA, USA) and a 1% penicillin/streptomycin (Thermo Fisher Scientific, Waltham, MA, USA) solution in humidified 95% air and 5% CO_2_ at 37 °C. For LDH, MTT and real-time PCRs, HEK-293 cells were seeded into 96-well plates (Sarstedt, Nümbrecht, Germany) at a density of 10^4^ cells/well (*n* = 5–6 wells/treatment group) and treated with LPS (1 µg/mL; *Escherichia coli* serotype 0111:B4; Merck, Kenilworth, NJ, USA), or H_2_O_2_ (100 µM) and Compound 23 (Compound-23) (0.1 µM, CAS: 724737-74-0) for 24 h.

### 2.2. Human Peripheral Blood Mononuclear Cells

Isolation and culture of human PBMCs were approved by Semmelweis University Regional and Institutional Committee of Science and Research Ethics (31224-5/2017/EKU). PBMCs from a healthy adult donor were isolated by density gradient centrifugation using Histopaque-1077 (Merck, Kenilworth, NJ, USA). After isolation, the cells were placed into Roswell Park Memorial Institute RPMI 1640 media (Thermo Fisher Scientific, Waltham, MA, USA) supplemented with 10% FBS and 1% penicillin/streptomycin solution in humidified 95% air and 5% CO_2_ at 37 °C. For real-time PCRs, PBMCs were seeded into 96-well plates at a density of 10^4^ cells/well (*n* = 5–6 wells/treatment group) in RPMI medium and treated with LPS (0.01 ng/mL), or H_2_O_2_ (100 µM) and Compound-23 (0.1 µM) for 24 h.

### 2.3. Animal Model

All experiments were approved by the institutional committee on animal welfare (PE/EA/696-8/2020). The experiments were performed on 7–8-week-old male C57Bl/6J mice (Charles River Laboratories, Germany), (*n* = 3–7 in each group). The animals were housed in a temperature-controlled (22 ± 1 °C) room with alternating light and dark cycles and had free access to standard chow and water. Acute kidney injury was caused by sepsis induction via a single intraperitoneal administration of LPS (5 mg/kg, *Escherichia coli* serotype 0111:B4; Merck, Kenilworth, NJ, USA) together or without Compound-23 (20 mg/kg) treatment. The dose of LPS treatment was determined based on previous literature data [23,24,25,26,27]. The control groups received an equal volume of vehicle or Compound-23 alone intraperitoneally. The experiment was terminated 24 h after the treatments and the mice were euthanized. Thereafter, blood samples and kidneys were harvested for further histological and molecular biological analyses. As a preliminary investigation, we used LPS at 10 mg/kg and Compound-23 at 10 mg/kg (Appendix A).

### 2.4. Immunofluorescent Staining

HEK-293 cells and PBMCs were seeded into 4-well chambers (Sarstedt, Nümbrecht, Germany) at a density of 6 × 10^4^ cells/well for 24 h. The fresh frozen OCT (Tissue-Tek O.C.T, Quiagen, Venlo, The Netherlands) embedded renal samples were cut into 4 µm sections. After fixation by methanol, chambers and tissue sections were first incubated with DJ-1 specific primary antibody (ab18257, 1:1000, Abcam, Cambridge, MA, USA) and then with Alexa Fluor 488^®^ conjugated secondary antibody (A21206, 1:1000, Thermo Fisher Scientific, USA). The nuclei were stained with Hoechst 33,342 (Merck, Kenilworth, NJ, USA). Finally, the slides were cover-slipped with ProLongTM Gold Antifade reagent (Thermo Fisher Scientific, USA). To visualize the stained sections, an Olympus IX81 microscope system was used (Olympus Corporation, Tokyo, Japan).

### 2.5. Protein Isolation and Western Blot

Kidney samples were homogenized in lysis buffer containing 1% Triton X-100, 1 mM EGTA, 5 mM NaF, 100 mM Tris, 1 mM phenylmethylsulfonyl fluoride, 1% Protease Inhibitor Cocktail and 10 mM Na3VO4 (pH 7.4; each substance was obtained from Merck, Kenilworth, NJ, USA). Protein concentration was determined in triplicates by a detergent-compatible protein assay (Bio-Rad, Hercules, CA, USA). The samples were denatured in Laemmli buffer at 95 °C for 5 min, and 20 μg protein/lane was loaded and separated on 4–20% gradient SDS polyacrylamide gel (Bio-Rad, Hercules, CA, USA), and transferred to nitrocellulose membranes (Bio-Rad, Hercules, CA, USA). The membranes were blocked with 5% non-fat milk. Thereafter, the membranes were incubated with antibodies specific for DJ-1 (ab18257; rabbit, 1:1000, Abcam, Cambridge, MA, USA) and GAPDH (sc-47724; Santa Cruz Biotechnology, Dallas, TX, USA). Thereafter, appropriate HRP-conjugated secondary antibodies (A0545, Merck, Kenilworth, NJ, USA; G21040, Invitrogen, Thermo Fisher Scientific, USA) were used. Bands of interest were detected using enhanced chemiluminescence detection (Western Blotting Luminol Reagent, GE Healthcare, Waukesha, WI, USA) and quantified by densitometry (VersaDoc, Quantity One Analysis v4.6.9 software; Bio-Rad, Hercules, CA, USA) as integrated optical density after background subtraction. Relative protein levels were determined by comparison with GAPDH as internal control. Data were normalized and presented as the ratio of control values.

### 2.6. RNA Isolation and cDNA Synthesis

Total RNA was isolated from mouse kidney samples, HEK-293, and PBMC cells using Total RNA Mini Kit (Geneaid Biotech Ltd., New Taipei City, Taiwan) following the manufacturer’s instructions. The concentration and quality of the isolated RNA were determined by a DeNovix DS-11 spectrophotometer (DeNovix Inc., Wilmington, DE, USA). A total of 500 ng RNA was reverse-transcribed using Maxima First Strand cDNA Synthesis Kit for RT-qPCR (Thermo Fisher Scientific, USA) to generate first-stranded cDNA.

### 2.7. Real-Time Polymerase Chain Reaction (PCR)

The mRNA expression of the target molecules was determined by real-time PCR using SYBR Green PCR master mix on a Light Cycler 96 system (Roche Diagnostics, Mannheim, Germany). The nucleotide sequences and species’ specificity of the applied primer pairs and the lengths of the resulting PCR products are shown in Table 1. The relative mRNA expressions were analyzed by Light-Cycler 96 software 1.1 and determined by comparison with GAPDH as an internal control using the ΔΔCt method. Data were normalized and presented as the ratio of their control values.

### 2.8. LDH Cytotoxicity Assay

The extent of cell death was determined by a colorimetric method, based on the lactate dehydrogenase (LDH) enzyme activity in the supernatant, released from damaged cells. Equal volumes (40 μL) of aspired media were mixed in a sterile 96-well plate with LDH reagent, containing 109 mM lactate, 3.3 mM ß-nicotinamide-adenine-dinucleotide-hydrate (N7004), 2.2 U/mL diaphorase (D2197), 3 mM TRIS, 30 mM HEPES, 10 mM NaCl, 350 μM thiazolyl blue tetrazolium bromide (all reagents were purchased from Merck, Kenilworth, NJ, USA), and then incubated at 37 °C for 1 h. Absorbance was recorded at 570 nm and 690 nm as background in a CLARIOstar microplate reader microplate reader using SPECTROstar Nano MARS v3.32 software (BMG Labtech, Ortenberg, Germany).

### 2.9. MTT Cell Viability Assay

The rate of cell proliferation was determined by a colorimetric method, based on the intracellular mitochondrial dehydrogenase activity of the attached cells. Then, 10 μL of MTT reagent, containing 5 mg/mL thiazolyl blue tetrazolium bromide (diluted in sterile H_2_O) was added into each well of the 96-well plate including cells and 100 μL of supernatant as well, followed by incubation at 37 °C for 4 h. Thereafter, the supernatants were removed from cells using a pipette, and the intracellular MTT crystals were dissolved by adding 100 μL 1:1 mixture of DMSO and ethanol (all reagents were purchased from Merck, Kenilworth, NJ, USA). Absorbance was recorded at 570 nm and 690 nm as background in a SPECTROstar Nano microplate reader using SPECTROstar Nano MARS v3.32 software (BMG Labtech, Ortenberg, Germany).

### 2.10. DCFDA Assay

Intracellular ROS accumulation was measured using 2′,7′-dichlorofluorescein diacetate (DCFDA, Merck, Kenilworth, NJ, USA) fluorescent dye. The cells were washed twice with PBS; thereafter, 50 μL of a DCFDA solution (5 μM in PBS) was added for each well and incubated for 30 min in a cell culture incubator. The fluorescence signal was measured for 30 min in every 5 min after induction of oxidative stress in the CLARIOstar microplate reader (λexc: 485 nm, λem: 535 nm) using MARS Data Analysis Software v4.01 (BMG Labtech, Ortenberg, Germany).

### 2.11. Measurement of Renal Parameters

Renal function parameters (serum creatinine and BUN) in mouse serum were determined by standard methods using commercially available kits on a Hitachi 912 chemistry analyzer (Roche Diagnostics, Mannheim, Germany).

### 2.12. Statistical Analysis

Statistical evaluation of data was performed using GraphPad Prism 8.01 software (GraphPad Software Inc., San Diego, CA, USA). After testing the normality with the Kolmogorov–Smirnov test, ordinary one-way ANOVA or the Kruskal–Wallis test were used to determine the differences among groups. The results are illustrated as mean ± SD with dots, representing individual values. *p* values of less than 0.05 were considered to be statistically significant. The number of elements (*n*) is indicated in each figure legend.

## 3. Results

### 3.1. DJ-1 Expression in Kidney, HEK-293 Cells and PBMCs

The presence of DJ-1 in the healthy renal tissue of mice, untreated human embryonic kidney cell line (HEK-293) with epithelial characteristics [28], and peripheral blood mononuclear cells (PBMCs) from healthy volunteers was investigated by immunofluorescent staining. DJ-1 was present in the glomeruli and epithelial cells of renal tubules both in the cortical and medullar regions of the kidney. We also demonstrated DJ-1 immunopositivity in the HEK-293 cells and PBMCs (Figure 1).

### 3.2. The Effect of DJ-1-Binding Compound-23 on the Oxidative Stress Response of Renal Cells

Our data showed that 1 μM Compound-23 decreased the viability of HEK-293 cells, while lower concentrations had no effect (Figure 2A). Therefore, in our in vitro studies, we used 0.1 μM Compound-23, which was the highest concentration that did not affect cell viability. We demonstrated that Compound-23 decreased the H_2_O_2_-treatment-induced reactive oxygen species (ROS) production of the cells (Figure 2C). Furthermore, our data showed that Compound-23 improved the viability of bacterial lipopolysaccharide LPS- or H_2_O_2_-treated HEK-293 cells (Figure 2D,E). In line with these data, Compound-23 treatment decreased the mRNA expression of the oxidative stress markers NAD(P)H quinone dehydrogenase 1 (*NQO1*) and glutamate-cysteine ligase catalytic subunit (*GCLC*) in the LPS-treated and that of *NQO1* in the H_2_O_2_-treated cells (Figure 2F–I). Moreover, our results showed that Compound-23 decreased the interleukin 6 (*IL6*) expression in LPS- and in H_2_O_2_-treated cells (Figure 2J,K).

### 3.3. The Effect of DJ-1-Binding Compound-23 on the Proinflammatory Cytokine Production of PBMCs

To examine the effect of Compound-23 on oxidative-stress-induced expression of proinflammatory *IL6,* PBMCs were exposed to H_2_O_2_ or LPS treatment. We found that Compound-23 decreased both the H_2_O_2_- and LPS-induced mRNA expression of *IL6* in PBMCs (Figure 3A,B).

### 3.4. The Effect of DJ-1-Binding Compound-23 on LSP-Induced Acute Kidney Injury (AKI)

To investigate the role of DJ-1 in the pathomechanism of LPS (5 mg/kg)-induced AKI, Compound-23 (20 mg/kg)-treated mice were utilized. No mortality was observed prior to the termination of the experiment. Our results demonstrated that although the amount of DJ-1 was unchanged at the investigated timepoint (Figure 4A), Compound-23 treatment decreased the serum levels of kidney injury markers, including creatinine and blood urea nitrogen (BUN) (Figure 4B,C). In accordance, we found that Compound-23 decreased the LPS-induced mRNA expression of renal damage markers kidney injury 1 (*Kim1*) and neutrophil gelatinase-associated lipocalin (*Ngal*) (Figure 4D,E). We also tested the effect of a higher dose of LPS (10 mg/kg) and a lower dose of Compound-23 (10 mg/kg) but observed no effect on *Kim1* and *Ngal* mRNA expression (Appendix A).

### 3.5. The Effect of DJ-1-Binding Compound-23 on the Renal Expression of Antioxidant and Proinflammatory Genes in the Mice Model of LPS-Induced Acute Kidney Injury (AKI)

Compound-23 (20 mg/kg) treatment reduced the mRNA expression of oxidative stress markers *Nqo1* and *Gclc* in the kidneys of mice with LPS (5 mg/kg)-induced AKI, compared to mice treated with LPS alone (Figure 5A,B). Similarly, LPS-induced mRNA expression of *Il6* was diminished by Compound-23 treatment (Figure 5C). We also tested the effect of a higher dose of LPS (10 mg/kg) and a lower dose of Compound-23 (10 mg/kg) but observed no effect on *Il6* mRNA expression (Appendix A).

## 4. Discussion

Sepsis is a life-threatening condition caused by the uncontrolled inflammatory response of the body to various exogenous microbial products, including bacteria, leading to the dysfunction of the affected organs [22]. In bacterial sepsis, LPS binds to Toll-like receptor 4 (TLR4), expressed mainly by immune and renal tubular epithelial cells, as well. TLR4 activation initiates signaling cascades leading to increased production of reactive oxygen species (ROS) and proinflammatory cytokines, which play a determinative role in tubular epithelial loss and impaired renal function [29,30]. AKI is one of the most common complications of sepsis, and it is associated with high mortality and poor outcomes due to the lack of targeted therapy.

In the present study, we demonstrated that Compound-23—a compound maintaining the DJ-1 activity—treatment diminished the LPS-induced oxidative stress and inflammation of the kidney and improved its functions. These data suggest that DJ-1 plays an important role against LPS-induced oxidative stress and inflammation and could serve as a potential therapeutic target for the treatment of AKI.

DJ-1 is a multifunctional protein whose antioxidant effect was widely demonstrated in neurodegenerative, malignant, and gastrointestinal diseases [4,5,11,21]. However, recently, the beneficial role of DJ-1 was suggested in renal diseases including polycystic kidney disease, diabetic nephropathy and renal fibrosis, as well. Despite the growing interest and the encouraging results about the renoprotective effect of DJ-1, our knowledge is still limited about its possible role in sepsis-induced AKI.

In this study, we demonstrated the presence of DJ-1 in the cortical and medullar regions of the kidney, in HEK-293 renal epithelial cells, and in human PBMCs (Figure 1). These observations raise the possibility of the role of DJ-1 in the pathomechanism of renal diseases, including sepsis-induced AKI.

In the next set of experiments, we investigated the effect of DJ-1 in HEK-293 cells exposed to bacterial LPS or oxidative stress. In these experiments, Compound-23, a DJ-1-binding compound developed by Kitamura et al., was used to protect DJ-1 from its overoxidation and thus to maintain its activity [19]. Accordingly, our result demonstrated that Compound-23 reduced the ROS production of H_2_O_2_-treated HEK-293 cells (Figure 2B). This result is in accordance with our recent study demonstrating that Compound-23 prevents the ROS production of colon epithelial cells [21]. Moreover, it was shown that silencing of DJ-1 expression in mouse proximal tubule cells increases their endogenous ROS production [31].

In the context of oxidative stress, we also investigated the effect of Compound-23 on cell death. Indeed, in line with our results above, we demonstrated that Compound-23 treatment improves the survival of the LPS- or H_2_O_2_-treated HEK-293 cells (Figure 2C,D). In accordance with our present observation, recently, the protective effect of Compound-23 against H_2_O_2_-induced cell death of duodenal epithelial cells was also demonstrated by our group [11,21]. These results may be closely related to DJ-1’s antioxidant functions.

DJ-1 can mediate both the formation and elimination of ROS through various mechanisms. Cellular injury impairs the integrity of mitochondrial respiratory complex I, which leads to leakage of the mitochondrial electron transport chain and the escaped electrons react with oxygen to form ROS. It has been demonstrated that DJ-1 binds to cytochrome C oxidase subunit NDUFA4, and NADH-ubiquinone oxidoreductase of mitochondrial respiratory complex I, thereby maintaining its stability. Accordingly, in DJ-1 gene-knockout cells, the activity of mitochondrial complex I is reduced, which ultimately leads to the increased formation of reactive oxygen radicals. DJ-1 can also scavenge the formed ROS directly on its redox-sensitive cysteine residues at amino acids C46, C53, and C106 [24]. The oxidation of these residues reduces the level of ROS by directly reacting with them. However, perhaps the most well demonstrated antioxidant effect of DJ-1 is its effect on the stabilization of antioxidant response inducing transcription NRF2 [13]. Indeed, previous studies have shown that DJ-1 stabilizes NFR2 by inhibiting Kelch-like ECH-associated protein-1-mediated NRF2 ubiquitination and degradation. Stabilized NRF2 can translocate to the nucleus, enhancing the transcription of antioxidant genes such as GCLC and NQO1 [14,15].

Therefore, in the next set of experiments, we investigated the effect of Compound-23 treatment on the expression of NRF2-regulated genes. Interestingly, we found that Compound-23 treatment decreased the expression of GCLC and NQO1 in LPS- or H_2_O_2_-treated HEK-293 cells, as well (Figure 2E–H). During oxidative stress, GCLC contributes to the antioxidant response by synthesizing glutathione, which prevents oxidative damage to proteins, lipids, and DNA by donating electrons to oxidized molecules [32]. Similarly, NQO1 helps directly detoxify reactive quinones and stabilizes antioxidant molecules [33]. Indeed, both NQO1 and GCLC are essential antioxidant enzymes that are produced to neutralize the effects of ROSs, and their expression correlates with the extent of oxidative stress [33,34]. Although initially our results may seem surprising, the low expression of GCLC and NQO1 observed in this experiment does not indicate that Compound-23 treatment decreased the expression of NRF2 related antioxidant genes. Instead, in accordance with our above finding about the effect of Compound-23 on the ROS production of the renal cells, it rather suggests that Compound-23 was able to maintain the activity of DJ-1 and thus reduce the formation of ROS, making it unnecessary to increase the expression of GCLC and NQO1.

In the next set of experiments, we tested the effect of Compound-23 on LPS- and ROS-induced proinflammatory cytokine production of PBMCs and HEK-293 cells, as well. Our results showed that Compound-23 treatment decreased both the LPS- or H_2_O_2_-induced *IL6* production of PBMCs (Figure 3A,B) and HEK-293 cells (Figure 2I,J). Previously, the role of DJ-1 in inflammation has been suggested in various organs. Indeed, our recent study showed that Compound-23 treatment decreased the intestinal IL-6 expression in the mice model of DSS-induced colitis [11]. Similarly, Zhang et al. found that DJ-1 deletion resulted in increased inflammatory cytokine production, including IL-1ß, IL-6, and monocyte chemoattractant protein 1 (MCP-1) in the colon of DSS-treated mice [35]. Inflammatory cytokine production is mainly mediated via the nuclear factor kappa B (NF-κB) pathway. Indeed, LPS-mediated TLR4 activation induces downstream signaling cascades, leading to the release of NF-κB from the inhibitor of nuclear factor kappa B alpha (IκBα) and to its translocation into the nucleus, resulting in the transactivation of the expression of proinflammatory genes, including IL1 and IL6 [36]. It has been demonstrated that DJ-1 facilitates the cytoplasmic interaction between IκBα and NF-κB, thereby inhibiting NF-κB translocation to the nucleus [37]. Moreover, DJ-1 prevents the generation of ROS, which are also well-known inducers of the activation of NF-κB pathway and cytokine production [38]. Taken together, these results demonstrate that Compound-23 can decrease the renal production of inflammatory cytokines during sepsis.

Finally, we examined the effect of DJ-1 in the mice model of LPS-induced AKI, which mimics the features of bacterial sepsis-induced AKI in humans. It is important to note that this model represents secondary AKI-induced by systemic inflammation following LPS treatment and differs from other types of AKI where direct renal ischemic or toxic injury leads to kidney damage. We found that LPS treatment increased serum creatinine and blood urea nitrogen (BUN) levels, indicating renal injury, which was diminished by the Compound-23 treatment of the mice (Figure 4B,C). Creatinine and BUN are traditionally used metabolic biomarkers of renal functions [39,40]. Their increased serum levels indicate declined filtration capacity, which is a hallmark of kidney injury [40]. Therefore, our results suggest that Compound-23 treatment improves the kidney functions of LPS-treated mice. In addition, we found decreased expression of kidney injury molecule-1 (*Kim1*) and neutrophil gelatinase-associated lipocalin (*Ngal*) in the kidney of the LPS and Compound-23-treated mice compared to that of LPS-treated ones (Figure 4D,E). The expression of KIM-1 is reportedly low in healthy kidneys but it increases in acute injury, where its tubular expression correlates with the extent of kidney damage [41]. In addition, NGAL has antioxidative properties and induction of its expression in sepsis is a compensatory response to ameliorate oxidative injury [42]. Therefore, their decreased expression indicates both less severe kidney injury and reduced oxidative stress. In accordance with the in vitro results, we found that Compound-23 treatment decreased the renal expression of oxidative stress markers including *Nqo1* and *Gclc*, as well as that of *Il6* in the LPS-treated mice (Figure 5A–C). These results suggest that Compound-23 treatment protected the kidneys during LPS-induced sepsis via antioxidant and anti-inflammatory mechanisms.

In summary, we demonstrated that Compound-23 treatment, which protects against the overoxidation of the Cys106 amino acid of DJ-1, preserves its activity and reduces the kidney injury of mice from the LPS-induced AKI. The protective effects of Compound-23 were associated with the antioxidant and anti-inflammatory effects of DJ-1 (Figure 6). Our data indicate that compounds that facilitate DJ-1 functions may have a therapeutic potential in the treatment of sepsis-induced AKI. It is important to note that, although these data are promising, further studies with larger sample sizes are needed to clarify the exact molecular mechanisms through which DJ-1-modulating compounds protect the kidneys against sepsis-induced injury.

## 5. Conclusions

Our study demonstrated that DJ-1 is a promising therapeutic target in sepsis-induced AKI. We found that pharmacological protection and activation of DJ-1 functions can reduce the oxidative stress and inflammation in the kidney during the course of sepsis-induced AKI.

## 6. Patents

Based on the work reported in this manuscript, an international patent application was filed titled “Benzamide derivatives as anti-inflammatory compounds and uses thereof”. Patent identification number: WO2021240187A1 [43].

## Figures and Tables

**Figure 1 antioxidants-14-00719-f001:**
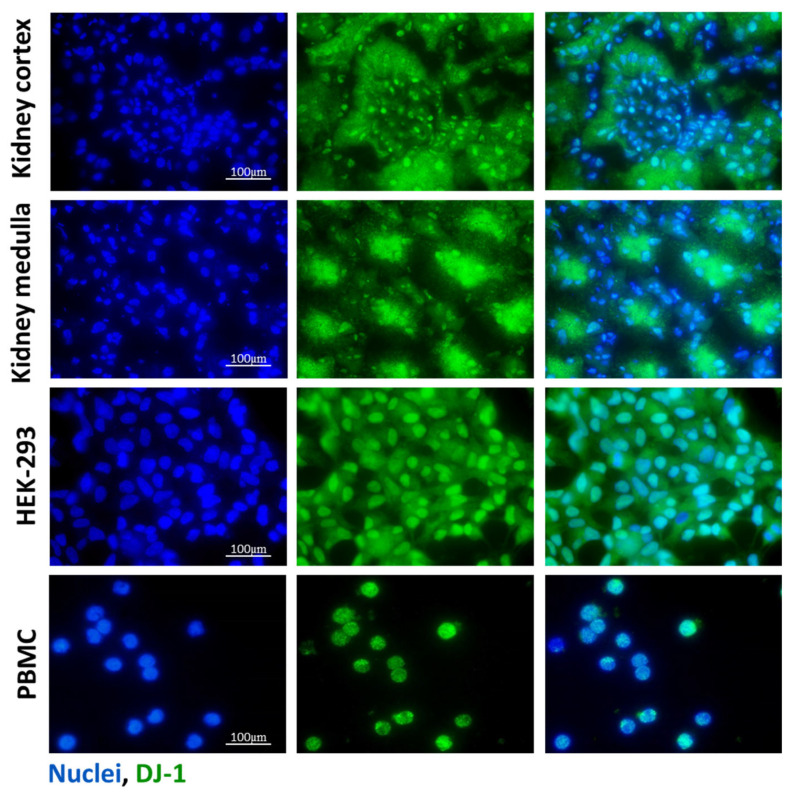
Presence of DJ-1 in the mice kidney samples, HEK-293 cells, and PBMCs. The presence of DJ-1 (green) was determined by immunofluorescent staining of the control kidney, human embryonic kidney cell line (HEK-293), and peripheral blood mononuclear cells (PBMCs). Cell nuclei were counterstained with Hoechst 33,342 (blue). Scale bar: 100 μm.

**Figure 2 antioxidants-14-00719-f002:**
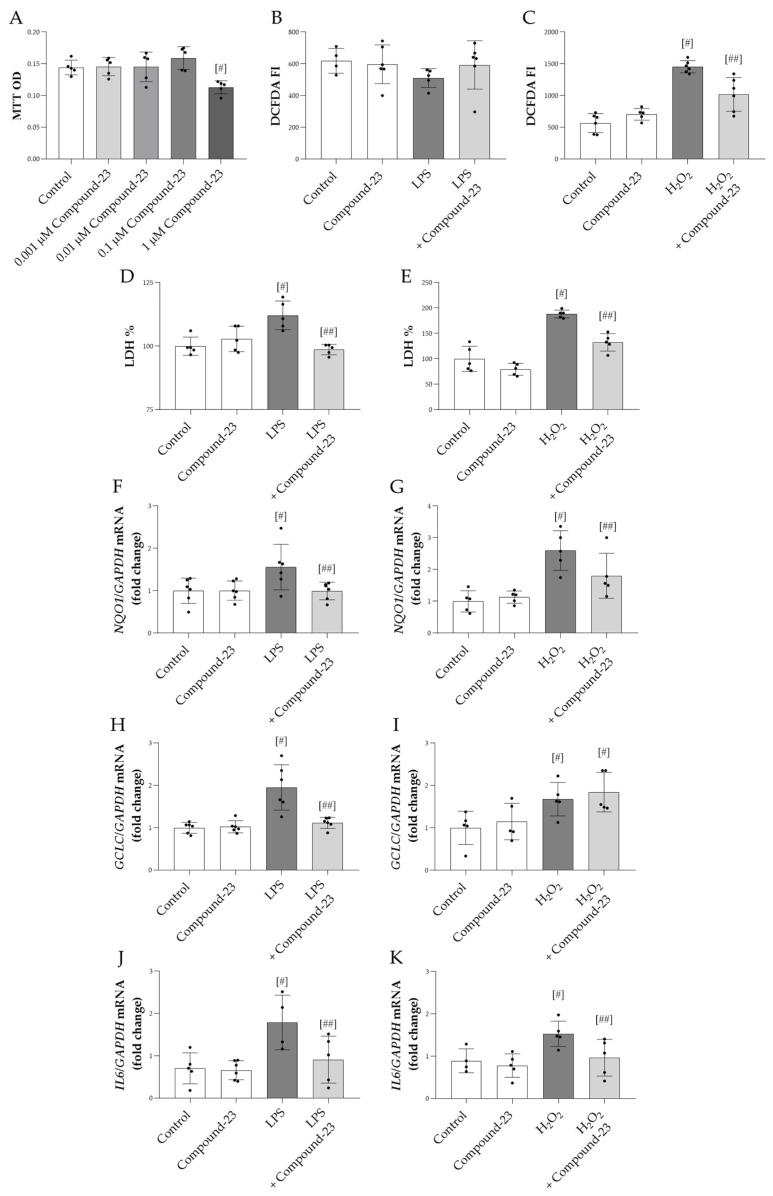
The effect of DJ-1-binding Compound-23 on oxidative-stress-induced cytotoxicity of HEK-293 cells. The effect of Compound-23 on the proliferation was investigated by MTT treated with Compound-23 (0.001 μM–1 μM) (**A**) for 24 h. HEK-293 cells were treated with LPS (1 µg/mL) (**B**,**D**,**F**,**H**,**J**) or H_2_O_2_ (100µM) (**C**,**E**,**G**,**I**,**K**) in the absence or presence of Compound-23 (0.1 μM) for 24 h. The effect of Compound-23 on intracellular reactive oxygen production was determined by DCFDA assay (**B**,**C**). The effect of Compound-23 on the viability of the HEK-293 cells was determined by LDH assay (**D**,**E**). The effect of Compound-23 on the mRNA expression of *NQO1*, *GCLC,* and *IL6* was determined by real-time PCR relative to with *GAPDH* expression (**F**–**K**). Data were normalized and presented as the ratio of the mean values of the control group. Values were expressed as mean ± SD. Dots represent individual values (*n* = 5–6). ^#^
*p* < 0.05 vs. control; ^##^
*p* < 0.05 vs. LPS or H_2_O_2_.

**Figure 3 antioxidants-14-00719-f003:**
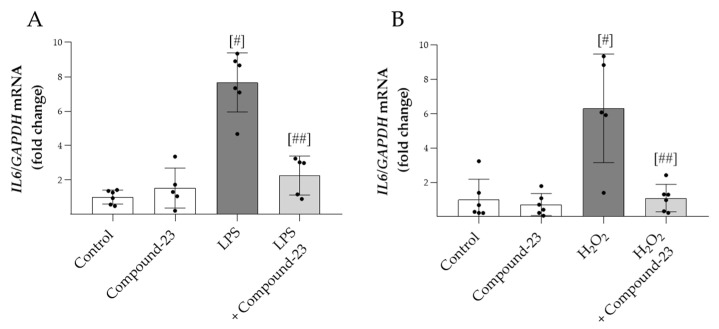
The effect of DJ-1-binding Compound-23 on proinflammatory cytokine production of PBMCs. PBMCs were treated with LPS (0.01 ng/mL) (**A**) or H_2_O_2_ (100 µM) (**B**) in the absence or presence of Compound-23 (0.1 μM) for 24 h. The effect of Compound-23 on the mRNA expression of *IL6* was determined by real-time PCR (*n* = 5–6) relative to *GAPDH* expression. Data were normalized and presented as the ratio of the mean values of the control group. ^#^
*p* < 0.05 vs. control; ^##^
*p* < 0.05 vs. LPS or H_2_O_2_.

**Figure 4 antioxidants-14-00719-f004:**
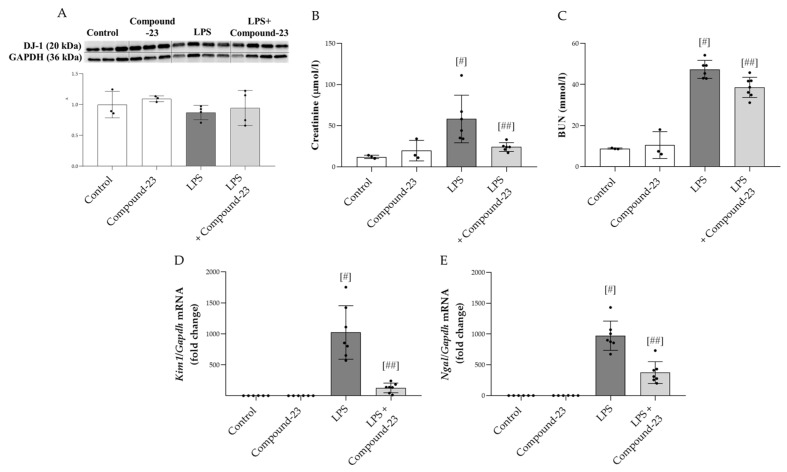
Effect of DJ-1-binding Compound-23 (20 mg/kg) on the LPS (5 mg/kg)-induced acute kidney injury (AKI). The protein level (**A**) of DJ-1 (20 kDa) in the kidney was determined by Western blot in comparison with GAPDH (36 kDa). The serum level of creatinine (**B**) and blood urea nitrogen (BUN) (**C**) were determined by enzyme assays. The renal mRNA expression of *Kim1* (**D**), *Ngal* (**E**) was determined by real-time PCR relative to *Gapdh* expression. Data were normalized and presented as the ratio of the mean values of the control group. Values were expressed as mean ± SD. Dots represent individual values in each group (*n* = 3–8); ^#^
*p* < 0.05 vs. Control; ^##^
*p* < 0.05 vs. LPS.

**Figure 5 antioxidants-14-00719-f005:**
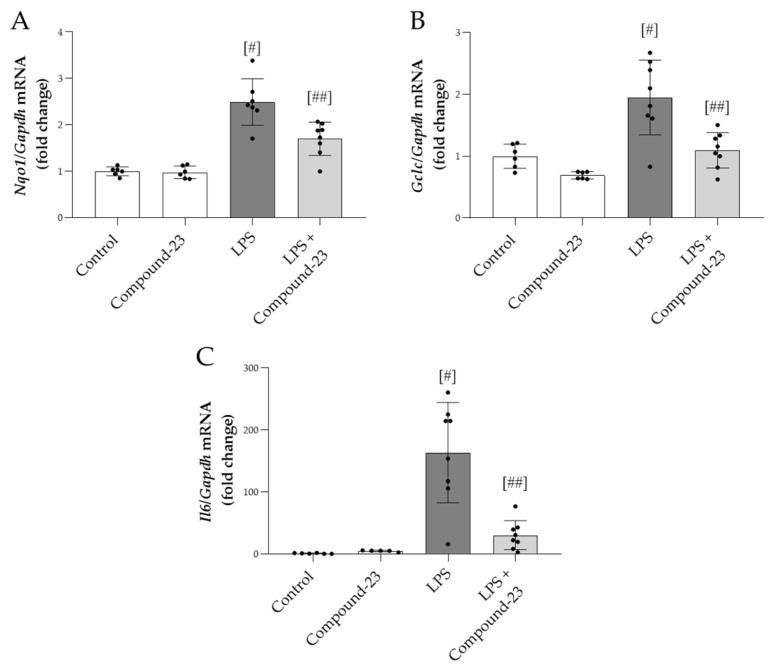
The effect of DJ-1-binding Compound-23 (20 mg/kg) on antioxidant and proinflammatory markers of kidney injury in the LPS (5 mg/kg)-induced acute kidney injury (AKI) model. The renal mRNA expression of *Nqo1* (**A**), *Gclc* (**B**), and *Il6* (**C**) was determined by real-time PCR relative to *Gapdh* expression. Data were normalized and presented as the ratio of the mean values of the control group. Values were expressed as mean ± SD. Dots represent individual values in each group (*n* = 6–8); ^#^
*p* < 0.05 vs. Control; ^##^
*p* < 0.05 vs. LPS.

**Figure 6 antioxidants-14-00719-f006:**
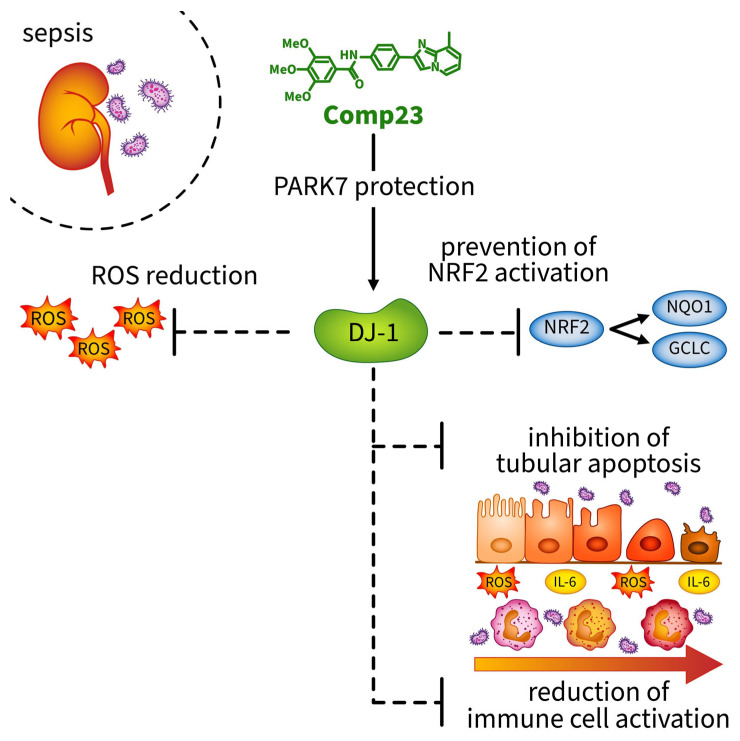
Compound-23 protects the antioxidant function of DJ-1 in the kidney, thereby reducing reactive oxygen species (ROS) formation. Reduction in ROS indirectly prevents the activation of the NRF2 pathway, as ROS are the primary inducers of NRF2 activation. Consequently, without NRF2 activation, the expression of the antioxidant genes NQO1 and GCLC remains lower. Moreover, treatment with Compound-23 reduces ROS-and LPS-induced apoptosis and IL-6 production in renal epithelial cells. The protective effect of Compound-23 on DJ-1 leads to less severe kidney damage in sepsis-induced acute kidney injury. Abbreviations: DJ-1: PARK7/Protein Deglycase DJ-1; ROS: reactive oxygen species; NRF2: NF-E2 related factor-2; NQO1: NAD(P)H quinone oxidoreductase 1 (NQO1); GCLC: γ-glutamylcysteine synthetase (GCLC); IL-6: interleukin-6.

**Table 1 antioxidants-14-00719-t001:** Nucleotide sequences of primer pairs were applied for the real-time polymerase chain reaction detection. Abbreviations: F: forward; R: reverse.

Organism	Gene	Primer Pairs
mouse	*Gapdh*	F: 5′-ATC TGA CGT GCC GCCTGGAGAAAC-3′
		R: 5′-CCCGGCATCGAAGGTGGAAGAGT-3′
mouse	*Kim1*	F: 5′-TCC AGG GAA GCC GCA GAA AAA-3′
		R: 5′-GGA AGG CAA CCA CGC TTA GAG ATG-3′
mouse	*Ngal*	F: 5′-GCC AGG CCC AGG ACT CAA CTC A-3′
		R: 5′-GTA CCACCT GCC CCG GAA CTGBAT-3′
mouse	*Nqo1*	F: 5′-TGG CCG AAC ACA AGA AGC-3′
		R: 5′-TGA ATC GGC CAG AGA ATG AC-3′
mouse	*Gclc*	F: 5′-GGA CTT TGA TGC GCC TCC TTC CTC TG-3′
		R: 5′-AAA CCC CAA CCA TCC GAC CCT CTG-3′
mouse	*Il6*	F: 5′-AAC CAC GGC CTT CCC TAC TTC A-3′
		R: 5′-TGC CAT TGC ACA ACT CTT TTC TCA-3′
human	*GAPDH*	F: 5′-AGC AAT GCC TCC TGC ACC ACC AA-3′
		R: 5′-GCG GCC ATC ACG CCA CAG TTT-3′
human	*NQO1*	F: 5′-CTG CTG CAG CGG CTT TGA AGA-3′
		R: 5′-GCC AGA ACA GAC TCG GCA GGA TAC-3′
human	*GCLC*	F: 5′-AAA AGT CCG GTT GGT CCT GTC TGG-3′
		R: 5′-GGC TGT CCT GGT GTC CCT TCA ATC-3′
human	*IL6*	F: 5′-AAA GAT GGC TGA AAA AGA TGG AT-3′
		R: 5′-CTC TGG CTT GTT CCT CAC TAC TCT-3′

## Data Availability

Data are contained within this article or in the Appendix A.

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
