# Peer review of "The DJ-1-Binding Compound Exerts a Protective Effect in Both In Vitro and In Vivo Models of Sepsis-Induced Acute Kidney Injury"

_antioxidants, 2025, doi:10.3390/antiox14060719_

Round 1

Reviewer 1 Report

The authors have addressed the topic of DJ-1: a therapeutic target in sepsis-induced acute kidney injury. The subject is very interesting, especially in the context of DJ-1's role as a potential target for the treatment of sepsis-induced acute kidney injury. The authors are requested to revise the manuscript in accordance with the reviewer’s comments:

  1. In the introduction of the paper, the authors should provide data on the expression and presence of DJ-1 in specific cells and tissues. These data are currently presented in the discussion section, but in my opinion, it would be valuable to include them in the introduction in order to present the current state of knowledge on this topic.
  2. Figure 1 should be included in the introduction and serve as a graphical representation of the conducted research.
  3. The figure is very good, but it should illustrate the impact of DJ-1 on the investigated pathways. In its current form, it is unclear what happens in response to COMP23 treatment—does it lead to a reduction in ROS levels? Does it enhance NRF2 activation? These effects should be clearly depicted.
  4. The introduction of the paper lacks a clear research objective and a detailed description of the planned experiments. Such a paragraph should be included either in the introduction or in the materials and methods section. The authors should clearly present the hypotheses of their study and explain how they intend to investigate and validate these assumptions. Otherwise, for instance while reading the paper, one gets the impression that the RT-PCR experiments lack purpose—there is no explanation of why the expression of the selected genes was analyzed.
  5. If the objective of the study is clearly defined, it will be easier to interpret and relate the obtained results in the discussion section. The discussion should be organized in a systematic manner—for example, by grouping the results according to DJ-1 expression, kidney injury assessed by creatinine levels, cell death (LDH release), effect of DJ-1 binding Comp23 on oxidative stress induced cytotoxicity and response to targeted therapy. In my opinion, the discussion should be revised and should focus exclusively on presenting the results in the context of data obtained by other authors. All other background information should be included in the introduction.

as above

Reviewer 2 Report

  • Clarification of Study Design
    Please clearly state in the title that the manuscript is based on both animal experiments and cell culture. This is essential for transparency and correct classification of the study.

  • Nomenclature Consistency
    The compound under investigation is referred to inconsistently throughout the manuscript. Please use a consistent term, preferably "Compound-23 (PARK7 / Protein Deglycase DJ-1)", across all text, figures, and legends to avoid confusion.

  • Quality of Immunofluorescence Imaging
    The immunofluorescence images appear blurry and poorly resolved, particularly in the sections showing tubular injury and PARK7 expression. High-resolution, focused images are essential for reliable interpretation. Please consider repeating these experiments or providing clearer images.

  • HEK293 Cell Morphology
    In Figure 1, HEK293 cells do not appear healthy. The cell morphology suggests stress or damage, independent of the experimental treatment. Please include representative images from untreated control cells, and confirm cell viability, ideally using additional methods (e.g., MTT or LDH assays).

  • Group Size Justification
    The use of 3 to 7 animals per group is not sufficiently justified. This group size appears underpowered, especially for detecting subtle biological differences in a heterogeneous model like LPS-induced AKI. A power analysis and justification for the sample size should be provided.

  • LPS and Compound Dosing
    The rationale for using 5 mg/kg of LPS should be clearly stated, as this dose is relatively high and may cause systemic shock in C57BL/6J mice. Similarly, the origin of the 20 mg/kg dose of Compound-23 is unclear. Please provide data or literature to support this dose selection. Were any animals euthanized or found dead before the 24-hour endpoint?

  • Survival Data and Clinical Scoring
    The manuscript would be significantly strengthened by including survival curves (Kaplan–Meier) and clinical scoring over time, especially since LPS can lead to early mortality. This is crucial to evaluate the true protective effect of Compound-23.

  • Classification of AKI
    It is important to explicitly state in the discussion that the model represents a secondary AKI, induced by systemic inflammation, and not primary ischemic or toxic AKI. This distinction has both mechanistic and translational relevance.

  • Toxicity Testing of Compound-23
    A toxicity assessment of Compound-23 should be included, particularly in HEK293 cells. Dose–response curves with at least 2–3 concentrations would help rule out direct cytotoxic effects and better define the therapeutic window.

  • Potential Overinterpretation and Patent Filing
    The conclusion and outlook sections seem overly optimistic, especially in light of the modest sample sizes and limited mechanistic exploration. If a patent has been filed based on these data,  the authors should critically reflect on whether the current results justify translational claims.

Line 2: please specify the title

Line 15; pleace specify compound 23

Line 78: please indicate incubation time 

Line 86: please explain animal number

Line 187 inconsistent nomemclature 

Line 205: incubation time 

Line 219 incubation time 

Line 369: limitations should be mentioned

Round 2

Reviewer 1 Report

Now is well done!

.